# Cathepsin Release from Lysosomes Promotes Endocytosis of *Clostridium perfringens* Iota-Toxin

**DOI:** 10.3390/toxins13100721

**Published:** 2021-10-12

**Authors:** Masahiro Nagahama, Keiko Kobayashi, Masaya Takehara

**Affiliations:** Department of Microbiology, Faculty of Pharmaceutical Sciences, Tokushima Bunri University, Tokushima 770-8514, Yamashiro, Japan; kobakei@ph.bunri-u.ac.jp (K.K.); mtakehara@ph.bunri-u.ac.jp (M.T.)

**Keywords:** *C. perfringens* iota-toxin, endocytosis, cathepsin

## Abstract

Iota-toxin from *Clostridium perfringens* type E is a binary toxin composed of two independent proteins: actin-ADP-ribosylating enzyme component, iota-a (Ia), and binding component, iota-b (Ib). Ib binds to target cell receptors and mediates the internalization of Ia into the cytoplasm. Extracellular lysosomal enzyme acid sphingomyelinase (ASMase) was previously shown to facilitate the internalization of iota-toxin. In this study, we investigated how lysosomal cathepsin promotes the internalization of iota-toxin into target cells. Cysteine protease inhibitor E64 prevented the cytotoxicity caused by iota-toxin, but aspartate protease inhibitor pepstatin-A and serine protease inhibitor AEBSF did not. Knockdown of lysosomal cysteine protease cathepsins B and L decreased the toxin-induced cytotoxicity. E64 suppressed the Ib-induced ASMase activity in extracellular fluid, showing that the proteases play a role in ASMase activation. These results indicate that cathepsin B and L facilitate entry of iota-toxin via activation of ASMase.

## 1. Introduction

*Clostridium perfringens* type E produces iota-toxin, a binary actin-ADP-ribosylating toxin composed of two distinct immunological proteins involved in animal enterotoxemia [1,2,3,4,5,6]. The enzymatic component, iota-a (Ia), ADP-ribosylate Arg177 of actin, interferes with the formation of actin filaments within host cells [6,7,8]. The binding component, iota-b (Ib), binds to lipolysis-stimulated lipoprotein receptors (LSRs) and mediates cell internalization of Ia into target cells [6,9]. We previously reported that Ib binds to nonlipid raft microdomains at the cell surface and induces rapid depletion of ATP and cell necrosis in A431 (human epithelial) and A549 (human lung) cells [10]. The clostridial binary actin-ADP-ribosylating family encompasses iota-toxin, *Clostridium botulinum* C2 toxin, *Clostridium spiroforme* toxin and *Clostridium difficile* ADP-ribosyltransferase [1,2,3,4].

Ib triggers cell internalization of Ia. Ib interacts with LSRs, the host cell receptors, on host cell membranes, generates heptameric Ib in the lipid raft microdomain of cell membranes, and docks with Ia in MDCK cells [9,11,12,13]. The structure of Ia complexed with Ib oligomer was recently solved by cryo-electron microscopy (cryo-EM) [14]. The complex of iota-toxin and LSR is endocytosed and transported to early endosomes [13,15]. At acidic endosome pH, the Ib heptamer inserts into the endosomal membrane and forms a pore, which permits translocation of Ia into the cytosol, with the assistance of the intracellular chaperone Hsp90 [1,2,3,4,5,6]. Ia in the cytosol catalyzes ADP-ribosylation of monomeric G-actin at Arg-177 [7,8]. ADP-ribosylated actin does not polymerize into actin cytoskeleton (F-actin) due to steric hindrance. Furthermore, ADP-ribosylated actin is a well-defined actin-capping protein that inhibits polymerization of nonmodified actin [1,2,3,4,5,6]. Subsequently, inhibition of actin polymerization leads to cell rounding [1,2,3].

We previously reported that ASMase is essential in promoting the internalization of iota-toxin by endocytosis [16]. Ib-evoked Ca^2+^ influx into the cytosol promotes exocytosis of ASMase from lysosomes. Extracellularly released ASMase induces ceramide accumulation on the outer surface of the plasma membrane. Ceramide-rich membrane areas promote endocytic plasma membrane invagination and iota-toxin endocytosis [16]. We recently demonstrated that lysosomal cathepsin B is an important contributor to the internalization of *C. botulinum* C2 toxin [17]. Lysosomal cathepsins have been reported to play a role in the pore-forming toxin (PFT)-mediated plasma membrane lesion repair pathway [18,19,20]. However, little is known about the role of cathepsins in the internalization of iota-toxin into host cells. Here, we provide evidence for the involvement of cathepsins in iota-toxin endocytosis.

## 2. Results

### 2.1. Effect of Protease Inhibitors on Iota-Toxin-Induced Cytotoxicity

To investigate whether lysosomal proteases play a role in iota-toxin internalization, we evaluated the effects of various proteinase inhibitors on the cytotoxicity of iota-toxin. As shown in Figure 1A, pepstatin A (PepA), an inhibitor of aspartic protease, and 4-(2-aminoethyl)benzenesulfonylfluoride HCl (AEBSF), an inhibitor of serine protease, did not influence the cytotoxicity of iota-toxin, while E64, an inhibitor of cysteine protease, inhibited iota-toxin activity. Bromoenol lactone (BEL) inhibits calcium-dependent PLA_2_, which is involved in vesicle transport and blocks lysosomal exocytosis [21]. BEL also blocked toxin-caused cytotoxicity (Figure 1A). Next, we examined the influence of E64 on the internalization of iota-toxin. After MDCK cells were incubated with Ib at 37 °C for 30 min, Ib appeared as small cytoplasmic vesicles in the presence of dimethyl sulfoxide (DMSO) (Figure 1B), indicating that Ib was internalized via endocytosis. In contrast, in the presence of E64, Ib was detected in the plasma membrane (Figure 1B), showing that E64 inhibited Ib endocytosis. These results indicate that lysosomal cysteine proteases play a role in iota-toxin endocytosis.

### 2.2. Iota-Toxin-Induced Release of Lysosomal Enzymes

We previously reported that Ib promotes ASMase release from lysosomes via Ca^2+^ influx [16]. To assess whether Ib causes local exocytosis of lysosomes, the secretion of lysosomal beta-hexosaminidase (beta-Hex) in the extracellular milieu of Ib-incubated cells was evaluated. As shown in Figure 2A, Ib time-dependently induced the release of beta-Hex in the culture supernatant. In contrast, the release of beta-Hex induced by Ib was blocked by BEL, indicating that Ib causes lysosome exocytosis. Extracellular enzymes such as lysosomal ASMase and cathepsins were previously reported to facilitate endocytosis and membrane repair [18,19,20]. To determine the role of lysosomal cathepsins in the endocytosis of iota-toxin, we assessed the Ib-induced extracellular release of the cysteine proteases cathepsin B and L and the aspartic protease cathepsin D. When MDCK cells were incubated with Ib, the activity of cathepsin B, L, and D increased in the culture supernatant at 5 min and then gradually declined (Figure 2B). Incubation of the cells with BEL reduced the increased cathepsin activity induced by Ib (Figure 2C), demonstrating that Ib promotes the release of lysosomal cathepsins. We previously reported that ASMase facilitates the endocytosis of iota-toxin into host cells [16]. In contrast, lysosomal cysteine proteases were reported to be involved in the activation of extracellular ASMase [19]. To clarify this issue, we examined the effect of protease inhibitor on ASMase activity (Figure 2D). In the presence of vehicle (DMSO), Ib increased the activity of ASMase in culture supernatant fluids in a time-dependent manner, indicating that Ib induced the release of ASMase. In contrast, increased ASMase activity through Ib was blocked by E64, indicating that lysosomal cysteine proteases are required for the activation of ASMase.

### 2.3. Effects of Iota-Toxin on Cathepsin siRNA-Treated Cells

To confirm the role of cathepsins in iota-toxin endocytosis, siRNAs targeting cathepsins B, L, and D were utilized to reduce cathepsin expression levels. As shown in Figure 3A–C, immunoblot analysis confirmed that siRNA treatment of cathepsins B, L, and D decreased the expression of each enzyme protein. siRNA-induced depletion of cathepsin B or L decreased the cytotoxicity of iota-toxin, whereas cathepsin D depletion did not affect iota-toxin cytotoxicity. These results indicate that lysosomal cathepsins B and L play a role in iota-toxin internalization, in accordance with the finding that E64 inhibits the cytotoxicity of iota-toxin.

## 3. Discussion

In this study, we discovered that the extracellular activity of lysosomal protease cathepsins B and L plays a role in the internalization of iota-toxin. First, cysteine protease inhibitor E64 prevents both cytotoxicity and internalization of iota-toxin, suggesting that the lysosomal cysteine proteases may promote endocytosis of iota-toxin. Second, Ib leads to enhanced release of lysosomal cysteine proteases through exocytosis into the culture supernatant. Third, RNAi-mediated knockdown of cathepsin B or L decreases the cytotoxicity of iota-toxin. Fourth, lysosomal cysteine proteases induce the activation of lysosomal enzyme ASMase, which promotes the endocytosis of iota-toxin. Taken together, our findings demonstrate that cathepsin B or L released from iota-toxin-treated cells via lysosomal exocytosis plays a role in iota-toxin internalization.

Iota-toxin internalizes target cells via endocytosis and causes cell rounding activity [1,2,3,4,5,6]. LSR is a cellular receptor for iota-toxin [9]. Ib binds to LSR through the C-terminal domain of Ib and assembles into Ib oligomerization and pore formation in lipid rafts of plasma membrane [13]. Ib then accelerates Ca^2+^ influx and promotes the release of lysosomal enzymes, such as ASMase and cathepsins, by Ca^2+^-triggered lysosomal exocytosis [15,16]. ASMase hydrolyzes sphingomyelin into phosphorylcholine and ceramide domains, triggering a rapid form of endocytosis of iota-toxin [16]. It has been reported that Ca^2+^-triggered lysosomal exocytosis induced by PFT facilitates the endocytosis and repair of plasma membrane [22,23]. In this study, ASMase was activated by co-released cathepsins B and L derived from lysosomes. In contrast, secreted cathepsins B and L were shown to be involved in the clearance of extracellular matrix proteins in plasma membranes [19]. This clearance of cell surface area contributes to the rapid access of ASMase to the plasma membrane sphingomyelin. Our results indicate that the release of lysosomal cathepsins B and L induced by Ib plays a role in the activation of ASMase and accessibility to its substrate in the plasma membrane. However, although Ib induces the release of lysosomal cathepsin D, knockdown of cathepsin D by siRNA and the aspartic protease inhibitor AEBSF did not block iota-toxin cytotoxicity. We think that the internalization of iota-toxin is not affected by cathepsin D. On the other hand, cysteine cathepsins are capable of remodeling the extracellular matrix [24]. We cannot rule out the possibility that cathepsins could lead to cell rounding independently of iota toxin activity.

We previously reported that binding of Ib to LSR promotes endocytosis of the Ib-LSR complex, resulting in colocalization of Ib with LSR in endosomes [13]. Ib and LSR are then transported to lysosomes for degradation [13]. The C-terminal domain of Ib (Ib421-664), named angubindin-1, binds to LSR without causing cytotoxicity. Ib421-664 alters the localization of LSR and tricellulin in tricellular tight junctions (tTJs), resulting in their increased permeability [25]. Although Ib binds to LSR in plasma membrane lipid rafts, Ib421-664 binds to nonlipid rafts [12]. Ib421-664 and LSR complexes are internalized and delivered to lysosomes for degradation, accompanying the decrease of LSR. In contrast, Ib421-664 did not induce pore formation or Ca^2+^ influx [12], indicating that it cannot promote Ca^2+^-triggered lysosomal exocytosis. Previous reports indicated that LSR plays a role in the cellular uptake of triglyceride-rich low-density lipoproteins [26]. We propose two pathways for the transport of iota-toxin to host cells. First, iota-toxin causes Ca^2+^-triggered exocytosis of ASMase and cathepsins from lysosomes in lipid rafts to internalize in the target cell. Second, Ib binds to LSR in nonlipid rafts and is internalized in such a way as to suggest that LSR takes up lipoproteins.

## 4. Conclusions

We demonstrated the role of lysosomal cathepsins B and L in the internalization of iota-toxin. Iota-toxin causes the release of cathepsins B and L and ASMase via Ca^2+^-dependent exocytosis of lysosomes. Cathepsins B and L promote internalization of iota-toxin in conjunction with ASMase.

## 5. Materials and Methods

### 5.1. Materials

We obtained purified recombinant Ia, Ib, and anti-Ib antibody, as reported previously [11,12]. Pepstatin-A, *p*-nitrophenyl *N*-acetyl-*β*-*D*-glucosaminide, E64, 4-(2-aminoethyl)-benzenesulfonyl fluoride hydrochloride, protease inhibitor cocktail and bromoenol lactone, enhanced chemiluminescence (ECL) kits, and horseradish peroxidase-labeled anti-rabbit and anti-mouse IgG were purchased from Merck (Tokyo, Japan). Mouse anti-cathepsin L (E-5) antibody and anti-cathepsin D (C-5) antibody were obtained from Santa Cruz Biotechnology (Santa Cruz, CA, USA). Rabbit anti-cathepsin B (D1C7Y) antibody and anti-beta-actin antibody were purchased from Cell Signaling (Tokyo, Japan). Amplex Red Sphingomyelinase Assay Kit, Dulbecco’s modified Eagle’s medium (DMEM), 4′,6′-diamino-2-phenylindole (DAPI), Alexa Fluor 488 phalloidin conjugate, Alexa Fluor 568-conjugated goat anti-rabbit IgG, and Hanks’ balanced salt solution (HBSS) were obtained from Thermo Fisher Scientific (Tokyo, Japan).

### 5.2. Cell Culture and Toxicity Testing

MDCK cells were obtained from RIKEN BioResource Center (Tsukuba, Japan), and were maintained in DMEM supplemented with 10% fetal calf serum (FCS). The medium contained 2 mM glutamine, 100 μg/mL streptomycin and 100 U/mL penicillin (FCS-DMEM). Cells were cultured at 37 °C in a 5% CO_2_ atmosphere. For toxin cytotoxicity assays, cells were seeded on 48-well culture plates, reaching confluence 24 h later. Cells were then treated with the various inhibitors or toxin. Cytotoxicity, as evidenced by cell rounding, was assessed 4 h after toxin treatment, as described previously [10].

### 5.3. Beta-Hexosaminidase Assays

MDCK cells were incubated with iota-toxin at 37 °C. At various time points, an aliquot of cell media was collected and evaluated for beta-hexosaminidase activity, which was determined by incubation with *p*-nitrophenyl *N*-acetyl-*β*-*D*-glucosaminide at 37 °C and pH 4.4 for 1 h, as previously described [16]. Enzyme activity is represented as the percentage of total enzyme activity found in the media and lysate.

### 5.4. Determination of Acidic Sphingomyelinase Activity

After incubation of MDCK cells with iota-toxin, an aliquot of supernatant was added to lysis buffer solution (50 mM sodium acetate buffer (pH 5.0) containing 1% Triton X-100 and 1 mM EDTA) supplemented with proteinase inhibitor cocktail. ASMase activity was evaluated using an Amplex^TM^ Red Sphingomyelinase Assay Kit, according to previously reported methods [16].

### 5.5. SiRNA Silencing and Western Blot Analysis

siRNA oligonucleotides for cathepsins B, D, and L and control non-silencing siRNA were obtained from Qiagen. MDCK cells were treated with siRNAs (1 × 10^5^ cells plus 250 pmol siRNA) and electroporated into MDCK cells utilizing a Neon^TM^ transfection system (Invitrogen), according to the manufacturer’s recommended method. The transfected cells were seeded in 24-well microtiter dishes and cultured in antibiotic-free FCS-DMEM at 37 °C. The cells were used for experiments 48 h after transfection [8]. Western blots were used to determine expression levels of cathepsins B, D, and L in cell lysates. Protein samples were separated on 10% SDS-PAGE and blotted on a polyvinylidene fluoride (PVDF) membrane. Western blotting utilizing specific antibodies against cathepsins B, D, and L and beta-actin was carried out as described previously [16].

### 5.6. Immunofluorescence

The immunofluorescence technique was used as previously described [10]. Briefly, MDCK cells were treated with Ib for 30 min at 37 °C. Cells were fixed with 3% paraformaldehyde for 10 min and permeabilized with 0.1% Triton X-100 for 10 min, followed by blocking with blocking buffer (PBS containing 4% BSA). Cells were then stained with rabbit anti-Ib antibody in blocking buffer for 1h. After washing, secondary Alexa Fluor 568-conjugated anti-rabbit IgG in blocking buffer was added and incubated for 1 h. Other staining was performed using cell nuclei with DAPI and actin with Alexa Fluor 488-phalloidin. The cells were analyzed under a Nikon A1 confocal laser scanning microscope.

### 5.7. Statistical Analysis

Statistical analysis was conducted using EZR (Easy R) statistical software (Jichi Medical University, Saitama Medical Center) [27]. One-way analysis of variance (ANOVA) combined with Tukey’s test was used to determine differences among three or more groups. Significant differences between two groups were evaluated using the two-tailed Student’s *t*-test. Differences were considered to be significant at values of *P* < 0.01.

## Figures and Tables

**Figure 1 toxins-13-00721-f001:**
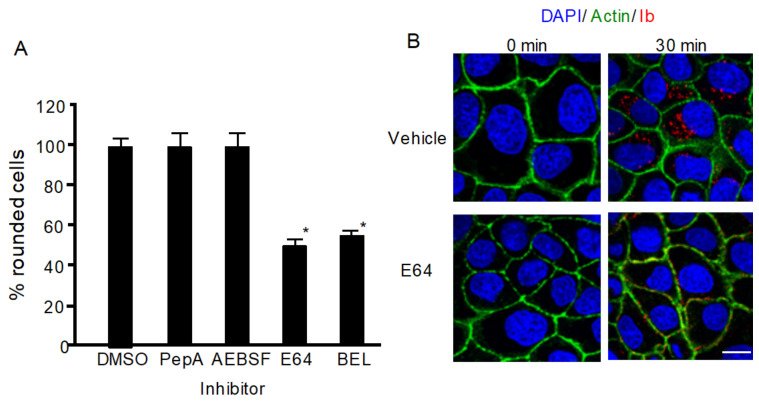
Effect of protease inhibitors on iota-toxin-caused cytotoxic activity. (**A**) MDCK cells were pretreated with 100 μM pepstatin-A (PepA), 100 μM E64, 100 μM AEBSF, 25 μM bromoenol lactone (BEL), or dimethyl sulfoxide (DMSO) (vehicle) at 37 °C for 1 h. Cells were then incubated with Ia (200 ng/mL) and Ib (500 ng/mL) at 37 °C for 4 h. About 100 MDCK cells were observed on microscopic pictures, and percentage of rounded cells was calculated. Values from five experiments are expressed as mean ± standard deviation (SD). One-way analysis of variance was utilized to assess significance. * *P* < 0.01, significant difference compared with vehicle (DMSO) plus iota-toxin. (**B**) MDCK cells were pretreated with 100 μM E64 or DMSO for 1 h at 37 °C. Cells were then treated with 500 ng/mL Ia and 1000 ng/mL Ib at 37 °C for 30 min. Cells were fixed and stained with an antibody against Ib, DAPI, and Alexa Fluor488-phallodin. Ib (red), actin (green), and nucleus (blue) were observed by confocal microscopy. Representative data of three studies are shown. Bar = 7.5 μm.

**Figure 2 toxins-13-00721-f002:**
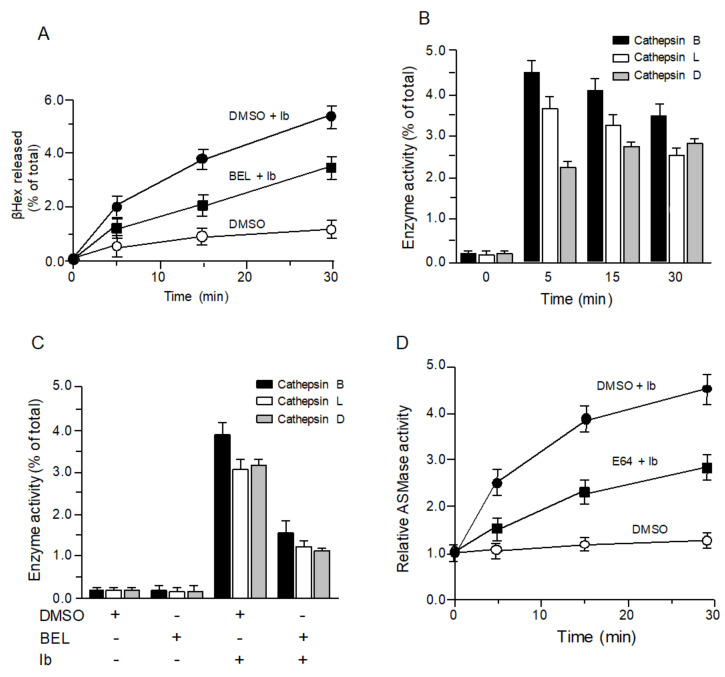
Iota-toxin causes exocytosis of lysosome. (**A**) MDCK cells preincubated with DMSO or 25 μM BEL were treated with Ib (500 ng/mL) at 37 °C at various time points. Aliquots of culture medium were assessed for beta-hexosaminidase (beta-Hex) activity, shown as percentage of total beta-Hex activity of cell lysates and supernatants. Data are mean of four studies ± SD. (**B**) MDCK cells were treated with Ib (500 ng/mL) at 37 °C. Aliquots of culture medium were assessed for cathepsin B, L, and D activity. Data indicate percentage of total cathepsin activity of cell lysates and supernatants. Data are mean of four studies ± SD. (**C**) MDCK cells pretreated with DMSO or 25 μM BEL were incubated with Ib (500 ng/mL) at 37 °C for 30 min. Aliquots of culture medium were assessed for cathepsin B, L, and D activity. Data indicate percentage of total cathepsin activity of cell lysates and supernatants. Data are mean of four studies ± SD. (**D**) MDCK cells pretreated with DMSO or 100 μM E64 were incubated with Ib (500 ng/mL) at 37 °C. Activity of acid sphingomyelinase (ASMase) in culture medium was analyzed as described in Materials and Methods. Untreated cells served as a control group, establishing a baseline level of 1.0. Data indicate percentages of values obtained from untreated controls. Data are mean of four studies ± standard deviation.

**Figure 3 toxins-13-00721-f003:**
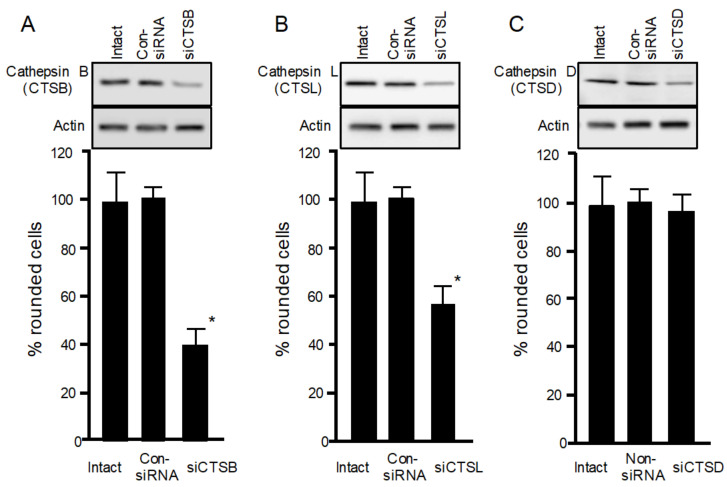
Role of cathepsins in iota-toxin-induced cytotoxicity in MDCK cells. Small interfering RNAs (siRNAs) were used to reduce (**A**) cathepsin B (siCTSB), (**B**) cathepsin L (siCTSL), and (**C**) cathepsin D (siCTsD). Non-silencing siRNA was utilized as control siRNA (Con-siRNA). Western blots were used to evaluate decreased levels of cathepsins B, L, and D. Data are representative of three independent experiments. siRNA-transfected cells were treated with Ia (200 ng/mL) and Ib (500 ng/mL) for 4 h at 37 °C. About 100 MDCK cells were observed on microscopic pictures, and percentage of rounded cells was calculated. Data from five experiments are expressed as mean ± SD. One-way analysis of variance was used to assess significance. * *P* < 0.01, significant difference compared with intact plus iota-toxin.

## Data Availability

Not applicable.

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
