# Peer review of "Cathepsin Release from Lysosomes Promotes Endocytosis of Clostridium perfringens Iota-Toxin"

_toxins, 2021, doi:10.3390/toxins13100721_

Round 1

Reviewer 1 Report

Clostridium perfingens iota toxin is a binary toxin with distinct catalytic (Ia) and cell-binding (Ib) components.  The toxin is internalized from the plasma membrane of a target cell, which eventually allows Ia to reach the cytosol where it ADP-ribosylates actin.  The resulting depolymerization of F-actin leads to cell rounding.  The authors have formulated a model in which Ib surface binding leads to lysosomal exocytosis and the release of a lysosomal enzyme, acid sphingomyelinase (ASMase), that enhances the endocytosis and resulting cellular activity of iota toxin.  The authors here show that lysosomal cathepsins also contribute to the cytotoxicity of iota toxin, most likely through activation of ASMase.  The experiments are straightforward, but the data interpretation fills in gaps that are not directly demonstrated by the experiments.

MAJOR POINTS

  1. The toxicity (cell rounding) assay of Figure 3 is used as a proxy for iota toxin internalization. Endocytosis of Ib is not directly demonstrated, but the data is interpreted as showing "cathepsin B and L play a role in iota-toxin internalization" (p. 4 lines 119-120).  Cathepsins are capable of remodeling the ECM, which could lead to cell rounding independently of iota toxin activity against F-actin.  If the authors want to state that cathepsins B and L play a role in iota toxin internalization (as opposed to iota toxin activity), they need to show this directly - as was done in the very nice confocal microscopy images of Figure 1B, which showed a general inhibition of cathepsin activity (as opposed to inhibition of a specific cathepsin) blocks Ib endocytosis.  If confocal is not used to directly document the lack of Ib internalization in cathepsin B/L-deficient cells, the authors need to revise their interpretation of this experiment to a determination that cathepsins B and L have roles in iota toxin activity rather than internalization (p. 4 lines 119-120 and 131-132; p. 5 lines 175-176).

  1. For the Western blots of Figure 3, were additional controls run to ensure the siRNA against cathepsin B did not result in the loss of cathepsin L and the siRNA against cathepsin L did not result in the loss of cathepsin B? In other words, are the authors sure their siRNA only affected the intended target and not other cathepsins?  If one siRNA affected both cathepsin B and L, then it is possible both cathepsins are not involved with iota toxin activity.  If the authors transfect cells with siRNA against both cathepsins, do they see a greater inhibitory effect than reported for cells transfected with siRNA against one individual cathepsin?  This would suggest multiple cathepsins contribute to the cytotoxicity of iota toxin.

  1. The Discussion states "the extracellular activities of lysosomal protease cathepsins B and L have roles in the internalization of iota-toxin" (p. 4 lines 131-132). Are the authors sure the cathepsins act upon ASMase in the extracellular environment?  If cathepsins and ASMase are both located in the lysosomes, then it is possible that the inhibition of cathepsin activity blocks ASMase activation in the lysosomes, before lysosomal exocytosis.  Why would the cathepsins only act on ASMase in the extracellular environment if they are located together in the lysosomes?

MINOR ISSUES

  1. In the Introduction, it is stated that "Ib binds to nonlipid raft microdomains at the cell surface" (p. 1 lines 24-25). Yet it is also stated that heptameric Ib is generated in the lipid raft microdomain of cell membranes (p. 1 lines 30-31).  Ib oligomerization in lipid rafts is also mentioned in the Discussion (p. 5 lines 143-144).  These apparent discrepancies should be clarified in the Introduction.  Does monomeric Ib first bind to its LSR receptor in nonraft membranes, which is followed by accumulation and oligomerization in lipid rafts?

  1. Please provide a reference for the use of BEL as an inhibitor of lysosomal exocytosis (p. 2 line 59). A one sentence explanation of how BEL prevents lysosomal exocytosis would also be helpful.

  1. The BEL and DMSO abbreviations should be defined upon their first use (p. 2 lines 59 and 62, respectively). They are currently defined in the legend to Figure 1, which comes after lines 59-62.

  1. The AEBSF and BEL abbreviations are defined in the Methods (p. 5 lines 183-184), but this is not necessary - they were already defined earlier in the Results section.

  1. In panel B of Figure 2, there appears to be a gap (empty space) between the cathepsin B and cathepsin L bars for the 30 min time point.

  1. For panels B and C of Figure 2, a y-axis label stating "% of total" would be more informative than the current "% of control".

  1. In the legend of Figure 2, there is almost no description of the conditions for panel B. More experimental detail is needed.  If the same experimental conditions were used for panels B and C, this should be explicitly stated.

Author Response

MAJOR POINTS

  1. The toxicity (cell rounding) assay of Figure 3 is used as a proxy for iota toxin internalization. Endocytosis of Ib is not directly demonstrated, but the data is interpreted as showing "cathepsin B and L play a role in iota-toxin internalization" (p. 4 lines 119-120). Cathepsins are capable of remodeling the ECM, which could lead to cell rounding independently of iota toxin activity against F-actin. If the authors want to state that cathepsins B and L play a role in iota toxin internalization (as opposed to iota toxin activity), they need to show this directly - as was done in the very nice confocal microscopy images of Figure 1B, which showed a general inhibition of cathepsin activity (as opposed to inhibition of a specific cathepsin) blocks Ib endocytosis. If confocal is not used to directly document the lack of Ib internalization in cathepsin B/L-deficient cells, the authors need to revise their interpretation of this experiment to a determination that cathepsins B and L have roles in iota toxin activity rather than internalization (p. 4 lines 119-120 and 131-132; p. 5 lines 175-176).

Answer: Cysteine cathepsins are involved in ECM remodeling (Forvic and Turk, BBA 1840, 2560 (2014)). Since Ib alone releases cathepsins by lysosomal exocytosis (Fig. 2A, 2B), we treated MDCK cells with Ib and observed cell morphological changes.However, Ib did not undergo any morphological changes in the cells. As cathepsin B and L are apparently not involved in ECM degradation, but rather in regulation of cell adhesion and invasion (Forvic and Turk, BBA 1840, 2560 (2014)), we think that cathepsin-induced changes in cell morphology are unlikely.On the other hand, the possibility of cell morphological changes due to cathepsins pointed out by the reviewer 1 cannot be completely ruled out. Therefore, the possibility of cell morphological changes due to cathepsin is also described in the discussion as follows.

p.5, line 157-159

On the other hand, cysteine cathepsins are capable of remodeling the extracellular matrix [24]. We cannot rule out the possibility that cathepsins could lead to cell rounding independently of iota toxin activity

  1. For the Western blots of Figure 3, were additional controls run to ensure the siRNA against cathepsin B did not result in the loss of cathepsin L and the siRNA against cathepsin L did not result in the loss of cathepsin B? In other words, are the authors sure their siRNA only affected the intended target and not other cathepsins? If one siRNA affected both cathepsin B and L, then it is possible both cathepsins are not involved with iota toxin activity. If the authors transfect cells with siRNA against both cathepsins, do they see a greater inhibitory effect than reported for cells transfected with siRNA against one individual cathepsin? This would suggest multiple cathepsins contribute to the cytotoxicity of iota toxin.

Answer: Experiments with cathepsin knockdown by siRNA have been reported by our recent report (Nagahama et al. Toxins 13, 272 (2021)), and same study has been made in other report (Castro-Gomes et al, Plos One 11, e0152583 (2016)). Neither report mentions the possibility that one siRNA may affect two enzymes. We think that one siRNA knocks down only its target molecule.In this report, individual knockdowns were performed to clarify the role of cathepsin B and L in the internalization of the toxin. As a result, it was found that knockdown of cathepsin B or L affects the internalization of the toxin. In addition, cysteine protease inhibitor E64 inhibited the toxin activity (Fig. 1A), andthe suppression of cathepsin B and L exocytosis by BEL inhibited the toxin activity (Fig. 1A and 2C). Taken together, we thnk that cathepsin B and L are involved in the internalization of toxins.

  1. The Discussion states "the extracellular activities of lysosomal protease cathepsins B and L have roles in the internalization of iota-toxin" (p. 4 lines 131-132). Are the authors sure the cathepsins act upon ASMase in the extracellular environment? If cathepsins and ASMase are both located in the lysosomes, then it is possible that the inhibition of cathepsin activity blocks ASMase activation in the lysosomes, before lysosomal exocytosis. Why would the cathepsins only act on ASMase in the extracellular environment if they are located together in the lysosomes?

Answer: In this study, the cysteine protease inhibitor E64 added to extracellular fluid inhibited ASMase activation by Ib. Beside, cathepsins may cleave cell surface proteins (Forvic and Turk, BBA 1840, 2560 (2014)) and contribute to membrane access of ASMase (as described in p.5, line 149-152).

MINOR ISSUES

  1. In the Introduction, it is stated that "Ib binds to nonlipid raft microdomains at the cell surface" (p. 1 lines 24-25). Yet it is also stated that heptameric Ib is generated in the lipid raft microdomain of cell membranes (p. 1 lines 30-31). Ib oligomerization in lipid rafts is also mentioned in the Discussion (p. 5 lines 143-144). These apparent discrepancies should be clarified in the Introduction. Does monomeric Ib first bind to its LSR receptor in nonraft membranes, which is followed by accumulation and oligomerization in lipid rafts?

Answer: Formation of toxin oligomer in cell membrane differs depending on the cell. In A431 and 549 cells, Ib oligomer binds to nonlipid raft microdomains at the cell surface (p. 1 lines 24-26) (Nagahama et al, Infect. Immun. 79, 4353 (2011)). On the other hand, in MDCK cells, Ib oligomer is generated in the lipid raft microdomain (p. 1 lines 30-31) (Nagahama et al, Infect. Immun. 72, 3267 (2004)). That is, iota toxin forms oligomer on both raft and non-raft, depending on the cell type. We think that monomeric Ib first bind to its LSR receptor in non-raft membranes. In A431 and 549 cells, Ib forms oligomer on non-rafts. In MDCK cells, Ib migrates and accumulates on rafts to form oligomer.

We added “in MDCK cells” in p. 1, line 31.

  1. Please provide a reference for the use of BEL as an inhibitor of lysosomal exocytosis (p. 2 line 59). A one sentence explanation of how BEL prevents lysosomal exocytosis would also be helpful.

Answer: OK. We added the explanation of BEL as follows:

In p.2, line59-60

Bromoenol lactone(BEL) inhibits calcium-dependent PLA2involved in vesicle transports and blocks lysosomal exocytosis (Fernandes et al., Front. Cell. Infect. Microbiol. 10, 39 (2020))[21].

  1. The BEL and DMSO abbreviations should be defined upon their first use (p. 2 lines 59 and 62, respectively). They are currently defined in the legend to Figure 1, which comes after lines 59-62.

Answer: OK. Wehave listed both abbreviations on the first page.

    In p. 2, line 59 :bromoenol lactone (BEL), line 63 : dimethyl sulfoxide (DMSO)

  1. The AEBSF and BEL abbreviations are defined in the Methods (p. 5 lines 183-184), but this is not necessary - they were already defined earlier in the Results section.

Answer: OK. We remove the abbreviations in the Method section.

In p. 5, line 184, 185.

5.In panel B of Figure 2, there appears to be a gap (empty space) between the cathepsin B and cathepsin L bars for the 30 min time point.

Answer: OK. We corrected Fig.2B.

  1. For panels B and C of Figure 2, a y-axis label stating "% of total" would be more informative than the current "% of control".

Answer: OK. We corrected y-axisof Fig.2B and 2C, as pointed out.

  1. In the legend of Figure 2, there is almost no description of the conditions for panel B. More experimental detail is needed. If the same experimental conditions were used for panels B and C, this should be explicitly stated.

Answer: OK. We added the explanation of Fig. 2B as follows:

In p. 3, Figure 2 legend, line 103-105.

(B) MDCK cells were treated with Ib (500 ng/ml) at 37 °C. Aliquots of culture medium were assessed for cathepsin B, L and D activities. Data indicate the percentage of the total cathepsin activity of cell lysates and supernatants. Data are the mean of four studies ± SD.

Reviewer 2 Report

This article clearly shows the results to support the author's claim.

To confirm the results more solid, I suggest the experiment shown in Figure 2 is conducted by cathepsin B and L knockdown.

Author Response

This article clearly shows the results to support the author's claim.

Thank you for your recommendation.

To confirm the results more solid, I suggest the experiment shown in Figure 2 is conducted by cathepsin B and L knockdown.

Answer : We thank you for your suggestions. We agree with you. We have demonstrated in Figure 2 that iota toxin induces lysosomal exocytosis to release cathepsins. Then, in Figure 3, we knocked down cathepsins to clarify the role of cathepsins in the toxin internalization. In this study, we think that the experimental results in Figure 2 and 3 sufficiently demonstrate the role of cathepsin in internalization of the toxin.

Reviewer 3 Report

In the present study, the authors investigated the roles of cathepsins in the internalization of Clostridium perfringens iota toxin internalization. The findings result in iota toxin b causing lysosomal exocytosis leading to cathepsin enhancing ASMase activity and thereby leading to an increasing cascade of iota toxin internalization. The results are overall convincing and well presented in the article. Some methodological details however need to be improved.

Major:

Cell viability measurements in Fig. 1A and Fig. 3 rely entirely on cell rounding, which scored by the investigator and therefore might be subjective. This method should be confirmed with a non-biased measurement such as ATP levels, MTT-test, or propidium iodide permeability.

In the experiment in Fig. 3, the investigators should do a cathepsin B+L double-knock-down to assess whether this confers additive protection against cytotoxicity.

Minor:

Despite the differences being clear, Figs. 2A, C, and D lack a statistical analysis of the differences between treated and untreated groups.

In Fig. 2B the cathepsin B bar at the 30 min time point is weirdly positioned away from the other measurements at the same timepoint.

Author Response

In the present study, the authors investigated the roles of cathepsins in the internalization of Clostridium perfringens iota toxin internalization. The findings result in iota toxin b causing lysosomal exocytosis leading to cathepsin enhancing ASMase activity and thereby leading to an increasing cascade of iota toxin internalization. The results are overall convincing and well presented in the article. Some methodological details however need to be improved.

Thank you for your recommendation.

Major:

Cell viability measurements in Fig. 1A and Fig. 3 rely entirely on cell rounding, which scored by the investigator and therefore might be subjective. This method should be confirmed with a non-biased measurement such as ATP levels, MTT-test, or propidium iodide permeability.

Answer. We thank you for your suggestions. Iota-toxin is a clostridial binary toxin with catalytic component (Ia) and binding component (Ib). Ib binds to a receptor and transfer Ia into cytosol, which Ia ADP-ribosylates actin. The resulting depolymerization of F-actin leads to cell rounding. Iota-toxin first rounds the cells. The rounding of the cells sharply reflects the action of iota-toxin. At this time, the morphology of the cells changes, but some rounded cells are dead, but some are alive. Some cells that are rounded by the action of toxins may recover and return over time. Cell death is not always quantitative. For clostridial binary toxin studies, cell rounding is widely used as an indicator of toxin activity.

In the experiment in Fig. 3, the investigators should do a cathepsin B+L double-knock-down to assess whether this confers additive protection against cytotoxicity.

Answer. We thank you for your suggestions. In Figure 3, we knocked down cathepsins to clarify the role of cathepsins in internalization of iota-toxin. The results indicate that cathepsins B and L are involved in internalization of iota toxin.That is, we think that the results clearly show the involvement of cathepsin.

Minor:

Despite the differences being clear, Figs. 2A, C, and D lack a statistical analysis of the differences between treated and untreated groups.

Answer. We thank you for your suggestions.As you pointed out, the difference in the Figure’s results is clear. We think that these figures are understandable. We have reported papers on similar experiments (Nagahama et al. Infect. Immun. 85, e00966-16 (2017), Nagahama et al. Toxins 10, 209 (2018)). In each report, there was no request for the results of the statistical analysis.

In Fig. 2B the cathepsin B bar at the 30 min time point is weirdly positioned away from the other measurements at the same timepoint.

Answer: OK. We corrected Fig.2B.

Round 2

Reviewer 1 Report

My critiques have been adequately addressed.

Reviewer 2 Report

I think this article presents sufficient data that support the author's claim.

Reviewer 3 Report

My comments have been addressed to my satisfaction.